# Surface Enhanced Infrared Absorption Studies of SiO₂–TiO₂–Ag Nanofibers: Effect of Silver Electrodeposition Time on the Amplification of Signals

Blanca Selenis Cabello-Ribota [1], Rurik Farías [2] and Simón Yobanny Reyes-López [1,*]

[1] Departamento de Ciencias Químico Biológicas, Instituto de Ciencias Biomédicas, Universidad Autónoma de Ciudad Juárez, Envolvente del PRONAF y Estocolmo s/n, Ciudad Juárez, Chih 32300, Mexico; al121451@alumnos.uacj.mx
[2] Instituto de Ingeniería y Tecnología, Universidad Autónoma de Ciudad Juárez, Ciudad Juárez, Chih 32584, Mexico; rurik.farias@uacj.mx
[*] Correspondence: simon.reyes@uacj.mx

**Abstract:** Surface Enhanced on Infrared Absorption (SEIRAS) and Raman Spectroscopy (SERS) are nondestructive analytic techniques used to detect low concentrations and recognize the fingerprints of molecules. The recognition of the absorption from samples by conventional infrared spectroscopy (IR) via Attenuated Total Reflection (ATR) is difficult for molecules with a low signal strength. However, developed structures with SERS and SEIRAS effect present problems such as high cost, low stability, and low compatibility. Research into new media to obtain greater amplification is largely based on the creation of nanoscale structures with symmetrical arrangements and reproducible distances, resulting in aggregates of nanoparticles that help generate hot spots which are active for amplification. The sol-gel and electrospinning method for the obtention of ceramics provides an alternative means by which to produce said substrates. Fibers of nanometric scale provide an increase of surface area which allows more contact to occur with analytes. Consequently, in this study, a silica-titania-silver nanostructured support that amplifies signal intensity for Raman and infrared spectroscopy was developed. The silica-titania support was developed by sol-gel and electrospinning techniques, and the as spun fibers were treated at 800 °C. Then, the ceramic fibrous membrane was placed on conductive indium tin oxide plastic to be doped with silver using an electroplating technique, varying the silver nitrate concentration (5, 10 and 20 mM), as well as electrodeposition times (1, 2, 5 and 10 min), with a constant voltage (1 V). Twelve different supports were obtained that showed amplification. The enhancement of infrared signals from pyridine and crystal violet molecules adsorbed on silica-titania-silver (SiO₂–TiO₂–Ag) nanofibers was studied in situ by Attenuated Total Reflection-Fourier Transformed Infrared Spectroscopy (ATR-FTIR). The highest amplification was obtained by the support doped at 10 min in a 10 mM concentration, with an amplification factor of 2.74 in the band localized at 3301 cm$^{-1}$. In Raman spectroscopy, the highest amplification factor was 27.03, on the support doped for 5 min at a concentration of 5 mM.

**Keywords:** attenuated total reflection infrared spectroscopy; ATR-FTIR; surface enhanced infrared absorption spectroscopy; SEIRAS; surface enhanced Raman spectroscopy; SERS nanoparticles; electrospinning; Sol-Gel

## 1. Introduction

Raman and ATR-FTIR spectroscopy are nondestructive analytic techniques used to study a broad range of diverse molecules under different conditions. SEIRAS is a modification of conventional infrared spectroscopy, in which the signal enhancement is exerted by the resonance plasmon of nanostructured metallic films [1]. SEIRAS permeates some of the limits of sensitivity of recognition produced by conventional ATR-FTIR. SEIRAS is a technique that uses the surface plasmon effect produced by the interaction of light

with metallic nanoparticles. The infrared absorption of molecules can be enhanced by metallic surfaces such as, copper, silver, and gold. Nanoparticles are deposited onto the surface, producing metal island films that give a localized enhancement of IR and Raman detection [1–3].

Infrared and Raman surface amplified spectroscopy is of great interest due to its ability to detect and sense ultralow concentrations (nM) and to recognize the "fingerprint" or characteristic spectrum of each substance. Therefore, it is applied in analytical chemistry and biological sciences. The detection and strength of the enhancement in SEIRAS and SERS depends on the synergistic effect of the metal nanostructure, surface morphology, and the nanostructure of the substrate support. The type of molecular binding with the metal is of great importance, as metallic island films close to separation show pronounced infrared enhancement [3–6].

The materials used as SEIRAS and SERS substrates are mainly metallic nanoparticles (gold, silver, or copper) or nanostructures (nanocapsules, nanowires, nanopoints, or nanoporous films), which have a high SEIRAS and SERS performance as a result of the localized surface resonance plasmon formed on metal due to the uniformity and repetitiveness of patterns [7,8] However, these structures show problems such as high cost and low stability and biocompatibility, in addition to not being suitable for creating large surfaces, preventing their large-scale introduction to the market [9]. Some of these problems have been solved by using semiconductors, such as $TiO_2$, to create active SEIRAS and SERS substrates, but the need to develop new materials that allow the SEIRAS and SERS effect to occur continues [10,11].

The reliable and reproducible manufacturing of suitable substrates represents a major challenge. Compared to processes using ultrahigh vacuum, wet chemical methods have the advantages of being less complex, inexpensive, and fast. Metallic nanoparticles are an attractive alternative for infrared enhancement. Molecules on the surfaces of noble metals or aggregates of nanoparticles frequently show a large Raman dispersion in certain sections. These sections are called "Hot spots"; they show amplification factors of $10^{14}$ and $10^{16}$ times and are found at the intersections between two metal particles due to plasmon coupling [12,13].

Chemical synthesis routes have an advantage for the preparation of metal nanoparticles or nanostructures because their properties change by varying their size, structure, and interparticle distance [14–16]. Sol-Gel and electro-spinning techniques make it possible to control the morphology of the composite on a nanometric scale, in a simple way and at room temperature, thus reducing production costs. In addition, the amount of silver particles electrodeposited on the fibers can be modified to control the number of hot spots in the support, thus causing the substrate to have a greater amplification effect [16–19]. Therefore, the development of a controlled structure composite, with a high number of hot spots, using a simple and low-cost method, will be of great benefit in areas such as diagnosis, analytical chemistry, pharmacy, biomedical, and photovoltaic technologies, as well as in studies on forensic and cultural heritage where SERS and SEIRAS are very useful techniques [4,20–22].

Not all molecules interact sufficiently with nanoparticles to perform SERS or SEIRAS measurements. Amplification can be observed if the molecules interact with nanostructures. As such, the energy of the interactions with the nanoparticles also influences the probability of observing amplification signals. One model is the extended Theory of Derjaguin, Landau, Verwey, Overbeek, where the interaction potential between two objects is a function of electrostatic, Van Der Waals forces and steric potential energies. Therefore, the sum of these interaction potentials is dependent on the separation distance of the particles. If the colloidal particles form aggregates or agglomerates, they remain in a state identical to that of the initial primary particles. The attractive forces in the short-range of Van der Waals bonds bring about the formation of clusters. It has been shown that the attraction potentials for silver and gold nanoparticles depend on the dielectric constants and the size of the metals, and that the magnitudes of these interaction potentials increase with decreasing

dimensions [23,24]. Smaller dimensions of a nanostructure, such as the tips of the peaks of gold nanostars, drive clustering behavior as well as amplification [23,24]. The signal enhancement is exerted by the resonance plasmon of nanostructured metallic structures of noble metals, like nanocapsules, nanowires, nanopoints, nanostars, and nanoporous films. Semiconductors, such as $TiO_2$ are used to create active SERS substrates. In this work, a support of silica-titania-silver fibers was obtained by a coaxial electro-spinning setup, using tetraethyl orthosilicate (TEOS) and titanium tetra-isopropoxide (TTIP) as precursors. Subsequently, the fibers were doped with silver particles by electrodeposition.

## 2. Materials and Methods

In this study, precursors of a metal-organic nature were used to produce ceramic materials. Tetraethylorthosilicate (Fluka, 99%) and titanium tetraisopropoxide (Sigma Aldrich, St. Louis, MO, USA, 97%) served as the silica and titania precursors. Polyvinylpyrrolidone (PVP) (Sigma Aldrich, 1300 kDa) was used to generate the electrospun gel compound. The solvent used for the different preparations was ethyl alcohol (Sigma Aldrich, purity: 99.5%) and deionized water. Acetic acid, deionized water, and hydrochloric acids from Sigma-Aldrich were used to formulate the precursor solutions. Silver nitrate (Sigma Aldrich, 99% purity) was used for the electrodeposition of silver. Precursor solutions were synthesized by the sol-gel method according to a modified method reported previously [3]. Molar ratios were 1:2:2:0.1 ($TEOS:CH_3CH_2OH:H_2O:HCl$) and 1:10:0:10 ($TTIP:CH_3CH_2OH:H_2O:CH_3COOH$). The solutions were mixed and homogenized with a PVP at 6% in ethanol at 1:1 before loading into a syringe for electrospinning in a coaxial setup. The processing parameters were voltage (13–14 kV), syringe drum-collector distance (12 cm), and flow (0.3 mL/h for the TTIP solution and 0.2 mL/h for the TEOS solution). The thermal treatment of the fibers consisted of drying at 50 and 100 °C for 24 and 4 h, with a subsequent sintering at 800 °C for 2 h with a heating ramp of 5 °C/min.

Doping with silver nanostructures was carried out using an electrolytic cell and a silver anode. The ceramic fibers were fixed on ITO plastic with a graphite adhesive tape and placed on the cathode of the cell. Subsequently, the electrodes were immersed in a silver nitrate solution at 60 °C, applying a voltage of 1 V during times of 1, 2, 5 and 10 min, in silver nitrate concentrations of 5, 10 and 20 mM.

The thermal evolution of weight loss and phase transitions from 20 to 1300 °C of precursor fibers were determined by Thermogravimetric Analysis (TGA), Differential Thermal Analysis (DTA) and Differential Scanning Calorimetry (DSC), using an SDT Q600 V20.9 Build 20 instrument (TA instrument, New Castle, DE, USA) using a heating ramp of 10 °C/min. Characterization by infrared spectroscopy was performed using Bruker FTIR ATR Platinum. The electrospun material was analyzed to evaluate its evolution through the heat treatment; therefore, samples were taken from the material prior to thermal and subsequent treatment of the (800 °C) twelve analyzed supports obtained after doping with silver. All samples were analyzed from 400 to 4000 $cm^{-1}$, with 4 $cm^{-1}$ resolution and 24 scans. WITec alpha 300 M+ confocal Raman spectrometer, 20× objective with a numerical aperture of 0.4, with a 532 nm laser and 10 s of integration time was used. Field Emission Scanning Electron microscopes SU5000 Hitachi and JEOL JSM-6010 PLUS/LA was used. Diffraction was employed to identify crystalline phases present in the fibers using the X'Pert PRO PANalytical, with Cu k$\alpha$ = 1.54, 20 kV, in the range of 2θ of 10 to 80°, a 2°/min scanning speed by the powder method. The amplification effect of nondoped and doped fibers was determined using 1 nM pyridine and 1 nM crystal violet solutions in deionized water, which were also used as the test samples. Then, 10 μL of pyridine (1 nM) and 10 μL of crystal violet were deposited on the fibers.

## 3. Results

The precursor solutions were prepared with some modifications to a methodology proposed previously [3], because in this work, 6% PVP (Mw 1,300,000) solutions were used for the electrospinning process. The resulting parameters were 12 cm from needle

to collector, a constant voltage of 13.5 kV, and a feeding rate of 0.3 and 0.2 mL/h for the titania and silica precursor solutions, respectively. The difference in polymer weight resulted in a more viscous solution, a characteristic that affects the process. Therefore, the electrospinning parameters were adjusted to obtain smooth fibers, free of beads or defects, as shown in Figure 1. The SEM micrographs in Figure 1a show that the fibers are oriented randomly, with mean diameters of $432 \pm 80$ nm, with smooth surface and without pores (Figure 1b). In Figure 1c, the EDX spectrum of the fibers at 25 °C indicates the presence of titanium (7.14%), silicon (1.64%), oxygen (27.91%), carbon (57.53%) and nitrogen (5.78%).

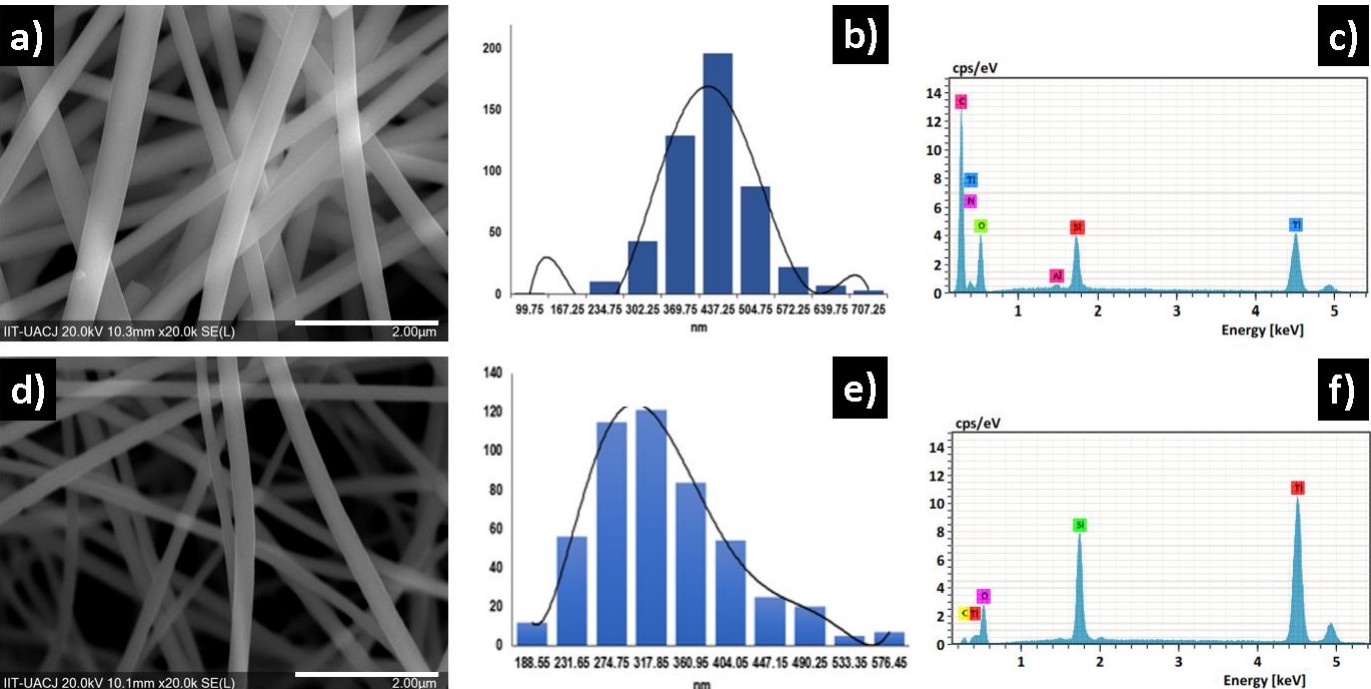

**Figure 1.** (**a**) SEM image, (**b**) Diameter size distribution and (**c**) EDX spectra of Fibers at 25 °C; (**d**) SEM image, (**e**) Diameter size distribution and (**f**) EDX spectra of Fibers treated at 800 °C.

The heat treatment at 800 °C was determined previously by evaluating the thermal evolution of the fibers. It was found that the chemical stability of the composite occurred at 800 °C, while at 1000 °C, the Si–O–Si chains in the fibers began to break. The micrographs in Figure 1d show that the morphology of the fibers remains, albeit with a smaller diameter of $280 \pm 90$ nm, because the organic matter was removed. The EDX spectra of fibers at 800 °C (Figure 1f) indicate that the main elements present were titanium (21.73%), silicon (8.03%) and oxygen (62.72%).

The as-spun fibers (PVP–SiO$_2$–TiO$_2$) were analyzed by infrared spectroscopy, obtaining the spectrum shown in Figure 2a. Most of the bands belong to functional groups in the polymer used (PVP). For example, the broad band located at 3450 cm$^{-1}$ is due to the stretching vibrations of the hydroxyl group (–OH), while the second band located at 2920 cm$^{-1}$ belongs to the stretching vibrations of the C–H bond of the –CH$_2$ group. The band that formed the symmetric and asymmetric stretching vibration of the C=O bond is found at 1657 cm$^{-1}$, the symmetric stretching vibration of the C–N bond of the pyridine ring was observed at 1495 cm$^{-1}$, and at around 1460 cm$^{-1}$, the scissoring vibration of the –CH$_2$ group and deformation vibration within the plane of the hydroxyl group are shown. The band located at 1374 cm$^{-1}$ belongs to the deformation vibration outside the plane of the C–H bond, and at 1435, 1288, and 1230 cm$^{-1}$, the bands that correspond to the deformation vibration outside the plane of the –CH$_2$ group were observed, although the latter two may also belong to the formation of an N–OH complex. Similarly, the bands located at 1172 and 845 cm$^{-1}$ are related to the symmetrical stretching vibration of the C–C

bond. At 740 cm$^{-1}$, the band belonging to the deformation vibration outside the plane of the C–H bonds of the ring was observed. Finally, the band located at 648 cm$^{-1}$ belongs to the deformation vibration of the C–N bond. The analysis of the as-spun PVP–SiO$_2$–TiO$_2$ fibers by ATR-FTIR results in bands located at 3450, 2920, 1657, 1460, 1374, 1288, 962 and 648 cm$^{-1}$, which agree with the bands of the PVP [22].

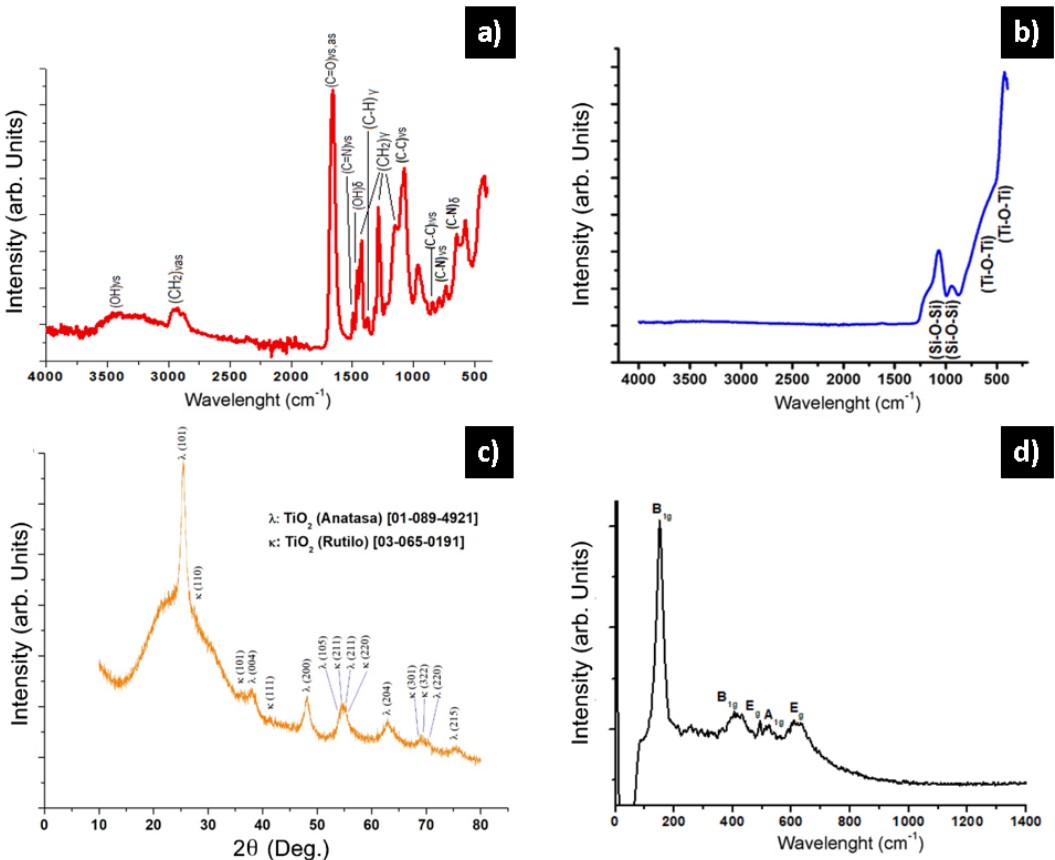

**Figure 2.** (**a**) IR spectra of fibers at 25 °C, (**b**) IR spectrum of fibers at 800 °C, (**c**) XRD of fibers at 800 °C and (**d**) Raman spectrum of fibers at 800 °C.

The polymer bands disappeared from the spectrum after heat treatment at 800 °C (Figure 2b), and only the bands belonging to the inorganic material of silica and titania were observed [3]. The infrared spectrum obtained from the SiO$_2$–TiO$_2$ ceramic fibers after heat treatment is shown in Figure 2b. The bands at 430 and 742 correspond to the stretching of the Ti–O bond, while those at 952 and 1080 cm$^{-1}$ belong to the stretching of the Si–O–Si bond. The X-Ray Diffraction characterization of ceramic fibers revealed the presence of the crystalline phases of titania: distinctive peaks of anatase phase were observed at 2θ 25.4°, 38.1°, 48.2°, 53.9°, 55.1°, 62.7°, 70.3° and 75.1°, which are associated to the planes (101), (004), (200), (105), (211), (204), (220) and (215) [01-089-4921]; and the peaks of the rutile phase were located at 27.5°, 35.6°, 41.3°, 54.42°, 55.5°, 68.9° and 69.4°, associated with the planes (110), (101), (111), (211), (220), (301) and (322) [03-065-0191]. The planes in Figure 2c are wide and with low intensity, which implies low crystallinity, with increased presence of the anatase phase. The diffractogram also shows a wide band between 15 and 35°, over which the anatase characteristic peak stands out, indicating the presence of an amorphous silica material contained in the fibers. When analyzing the silica-titania fibers by X-ray Diffraction, the crystalline phases anatase and rutile of titania were found at 800 °C. It would be expected to find only the rutile phase, because when thermal treatment is increased to more than 550 °C, the anatase phase gradually becomes rutile. However, in this case, the presence of silica prevents the transformation of anatase,

making it possible to find both phases [25,26]. Figure 2d shows the Raman spectrum of the $TiO_2$–$SiO_2$ fibers treated at 800 °C. Characteristic bands are identified for B1g vibrational modes at 409 and 526 $cm^{-1}$ for the anatase crystalline phase. The bands located at 409 and 611 $cm^{-1}$ correspond to the Eg and A1g movements of the rutile phase. The highest intensity band located at 151 $cm^{-1}$ is attributed to vibrational modes Eg and B1g present in both crystalline phases [27]. The formation of stable tetragonal anatase below 700 °C, and rutile tetragonal structure close to 700 °C, is known. The presence of both crystalline structures at temperatures up to 1200 °C was attributed to an interference of silica in rutile formation in the hybrid silica-titania composites. Therefore, it is possible to attribute the presence of both crystalline structures in the fibers treated at 800 °C to the interference of silica during the sintering of the composite.

The silica-titania fibers were doped with silver by an electrolytic process, varying the electrodeposition time and the concentration of silver nitrate solutions inside the cell, thus obtaining twelve different supports. Figure 3a shows the fibers deposited at a concentration of 5 mM with one minute of treatment. Only an isolated silver particle is observed in the micrograph, indicating that very little silver had been deposited. Figure 3b shows the micrographs obtained from the fibers doped for 2 min with a 5 mM concentration of silver nitrate. The 2 min treatment allowed a greater presence of particles and the formation of dendritic structures with average length of 4.42 ± 1.64 μm, and average width of 1.49 ± 0.49 μm. In addition, Figure 3b shows two crystals that emerged from the same nucleus, which contains ordered and well-defined (111) sliding systems. The micrographs obtained from the fibers with 5 min of treatment in the same concentration (Figure 3c) allow for the observation of the largest size reached by the dendrites, with an average length of 16.32 ± 0.91 μm and width of 2.24 ± 0.94 μm. Figure 3c shows how at longer time there is a greater amount of deposited silver. The dendrite grew along the fibers and not in isolated points with agglomerates giving a fiber with several silver nuclei. Figure 3d shows that after 10 min of treatment there is a greater presence of silver particles, but they are not homogeneously dispersed in the fibers. Instead, the particles are only located in certain areas. Figure 3d shows dendrites, almost spherical silver particles and a few crystals with no defined faces. There are also hexagonal silver crystals. The average dendrite size was 4.64 ± 1.55 μm, with a width of 2.01 ± 1.21 μm.

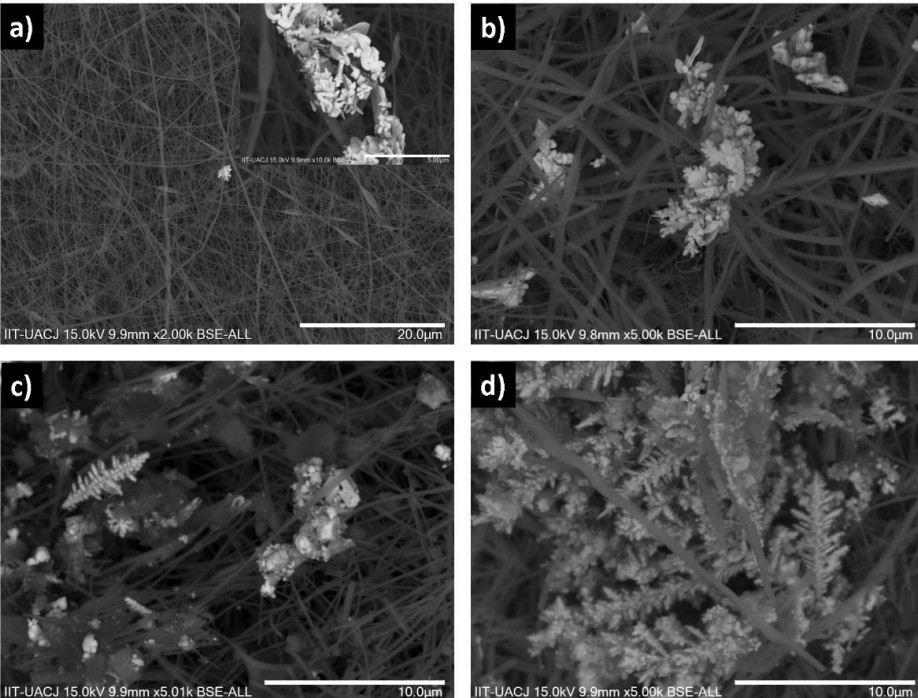

**Figure 3.** SEM micrographs of silver-doped fibers at 5 mM $AgNO_3$ during: (**a**) 1, (**b**) 2, (**c**) 5 and (**d**) 10 min.

Treatment of 1 min at a concentration of 10 mM led to the formation of defined dendritic structures. Figure 4a shows the average dendrite size in 9.11 ± 5.88 μm, with a width of 2.23 ± 1.43 μm. The EDX spectrum in the fibers confirmed the composition of silica-titania-silver. The 2 min electrolytic treatment revealed heterogeneous behavior regarding the position of the particles, as shown in Figure 4b, but with preferential longitudinal growth. At the tip of some dendrites, it is possible to observe how hexagonal silver crystals are formed. The average length of the dendrites was 6.55 ± 2.25 μm, with a width of 1.38 ± 0.58 μm. Figure 4c shows the fibers doped for five minutes at a concentration of 10 mM, approximately six silver hexagonal crystals were deposited on the silica-titania fibers. Silver dendritic structures have an average length of 3.70 ± 1.95 μm, and an average width of 0.92 ± 0.56 μm. Micrographs at these concentration and voltage conditions show branching nuclei, and dendritic growth is decreased by a preference for longitudinal growth over the fibers. The doped fibers obtained at 10 min of electrolytic treatment in Figure 4d show the formation of amorphous silver particles.

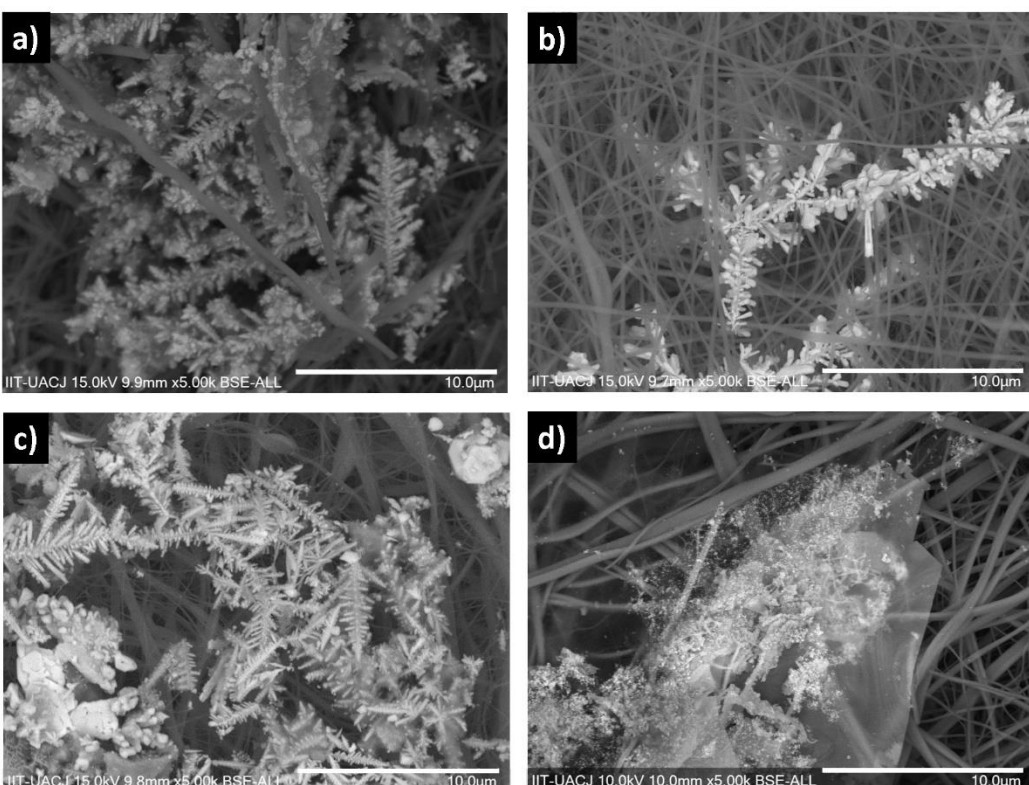

**Figure 4.** SEM micrographs of silver-doped fibers at 10 mM AgNO$_3$ during: (**a**) 1, (**b**) 2, (**c**) 5 and (**d**) 10 min.

The micrographs shown in Figure 5a are of the silica-titania fibers doped for 1 min at a concentration of 20 mM. Figure 5a shows a small amount of silver particles deposited with undefined crystals and in different zones; also, the particles are agglomerating and are hardly deposited. Figure 5b shows that despite the short treatment time, i.e., 2 min, there was dendrite formation, due to the high concentration of silver nitrate. Figure 5c shows the formation of wide dendrites and agglomerates of small and thin dendrites; the dendrites have an average length of 2.75 ± 0.92 μm and width of 0.70 ± 0.24 μm. Silica-titania-silver fibers obtained with 10 min of electroplating treatment at a concentration of 20 mM silver nitrate are shown with many dendrites, spherical particles, and silver hexagonal plates along the resulting fibers due to the high exposure time.

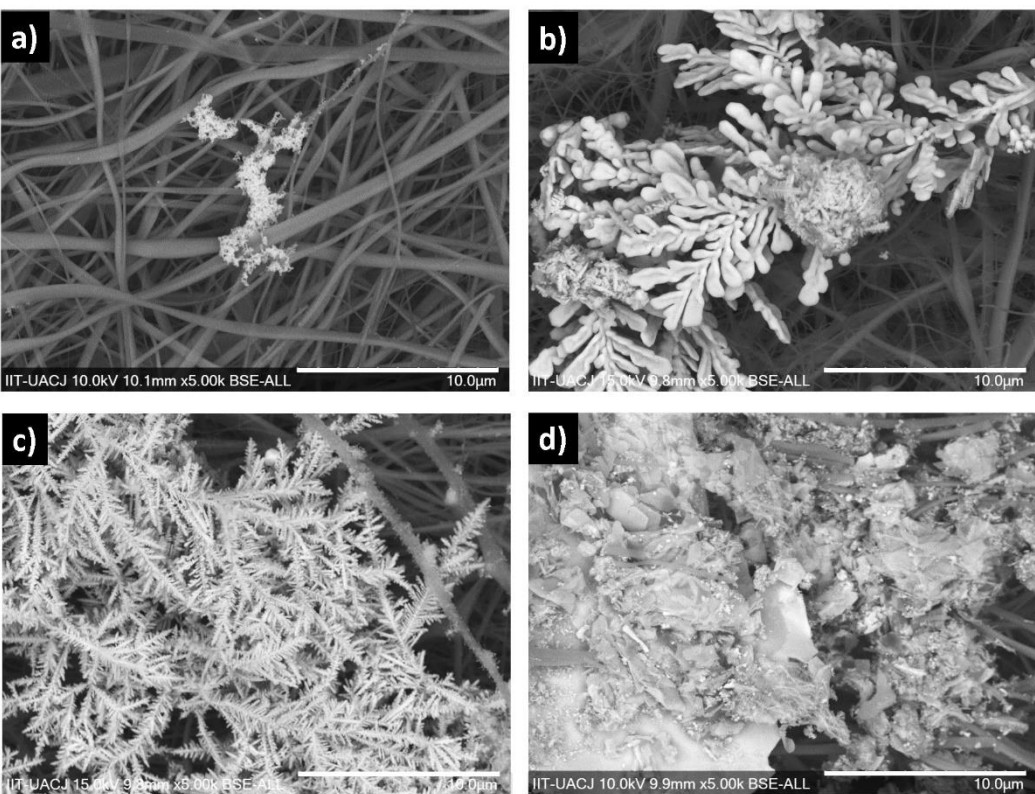

**Figure 5.** SEM micrographs of silver-doped fibers at 20 mM AgNO$_3$ during: (**a**) 1, (**b**) 2, (**c**) 5 and (**d**) 10 min.

The lowest concentration of silver nitrate used was 5 mM, and when applying voltage for 1 min, a small agglomeration of particles without defined crystalline faces was observed. After 2 min, there were already dendritic structures and crystals in which ordered sliding systems could be distinguished. At 5 min, most structures were dendrites that grew along the fibers, while after 10 min, almost spherical particles and hexagonal crystals could be observed. These structures were also observed by Yasnikov and Tsybuskina [28]. The behavior of the structures in the silver deposits is due to the fact that when applying the electric current for a short period of time, the nucleation phenomenon occurs, where the particles anchor themselves randomly and have a small growth "layer by layer". The electric current applied over a longer period stops favoring nucleation, and the growth of dendrites and stratified structures begins. Within two minutes these dendrites can still have amorphous areas, so they are somewhat thick. After 5 min there is an elongation of the trunk of the dendrites, as well as a compaction in this and in the branches, so its appearance is more defined. Finally, after 10 min, the dendrite morphology undergoes a transition in which the particles merge forming compact hexagonal plates and crystalline units [12,21,29,30].

*3.1. Evaluation of Signal Amplification in Infrared Spectroscopy*

In order to verify that the amplification effect occurs in bands of pyridine (Py) and not in water in infrared spectroscopy, a solution of concentrated pyridine (99.7%) alone in diamond crystal using the ATR technique and pyridine on the support was analyzed; the observed spectrum is shown in Figure 6. Bands located at 3078, 3033, 3023, 1580, 1482, 1435, 1213, 1147, 1068, 1029, 988, 746, 701, 676, 603, and 403 cm$^{-1}$ were observed, which belonged to different vibrational modes of pyridine, in addition to the bands located at 1080, 952, 742, and 430 cm$^{-1}$ that belonged to the ceramic support. The vibrational modes of the observed bands are shown in Table 1 and those of pyridine are illustrated in Figure 6 [30,31]. The infrared bands generated by the 1 nM pyridine solution are shown in Table 2 and Figure 7a,c,e. The first wide band with the most intense point between 3200 and 3500 cm$^{-1}$ was due to the stretching vibrations of the ring C–H bond. The intensity of the second band

was low, and it was observed at 2100 cm$^{-1}$, which is characteristic of a monosubstituted ring. The third band located at 1636 cm$^{-1}$ was formed by the stretching vibration of the C=N bond, and the fourth band over 500 cm$^{-1}$ was due to vibration within the plane by deformation of the ring. The spectra obtained from the pyridine on the supports show the bands of the Si–O and Ti–O bonds, as previously described, plus the bands located at 3200 and 1636 cm$^{-1}$. The silver doped supports in a concentration of 5, 10, and 20 mM AgNO$_3$ did not show any remarkable behavior, with all four supports yielding the same spectrum. However, in the supports doped at different times in the concentrations of 5, 10, and 20 mM, the spectra showed amplification of the bands for pyridine. The area of bands 3301 and 1636 cm$^{-1}$ of the supports doped at 5 mM yielded the data shown in Figure 7b, where it may be observed that the highest amplification for the support at 5 mM was for 2 min of electrodeposition. The area of these two bands for 10 and 20 mM in Figure 7d,e shows that the highest amplification between each concentration occurred in the support deposited for 5 min. The highest amplification for the twelve types of supports was observed at 10 min in a concentration of 10 mM AgNO$_3$.

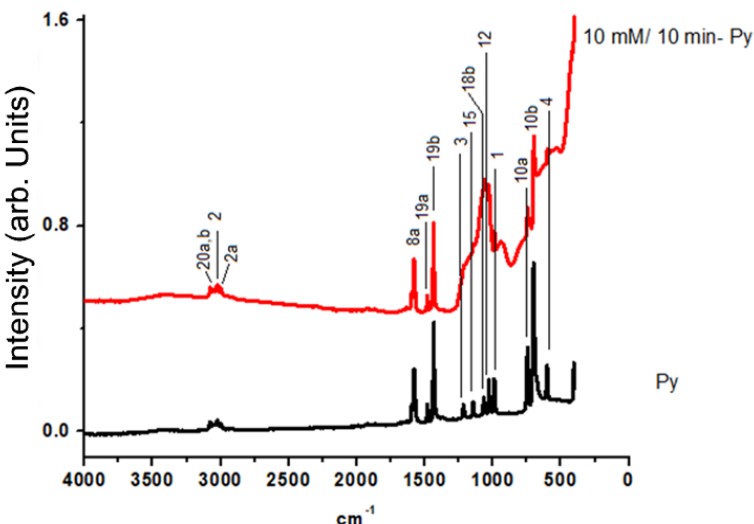

**Figure 6.** Infrared spectra of pyridine (Py) concentrated alone and pyridine on the support doped for 10 min in a concentration of 10 mM AgNO$_3$.

**Table 1.** Infrared spectrum bands obtained from Py concentrated alone and Py on the support doped for 10 min in a concentration of 10 mM AgNO$_3$ [3,30,31].

| Wavelength (cm$^{-1}$) | Band |
|---|---|
| 3078 | 20a, 20b |
| 3033 | 2 |
| 3023 | 2 |
| 1580 | 8a |
| 1482 | 19a |
| 1435 | 19b |
| 1213 | 3 |
| 1147 | 15 |
| 1068 | 18b |
| 1029 | 12 |
| 988 | 1 |
| 746 | 10a |
| 701 | 10b |
| 676 | 4 |
| 603 | 6a |
| 406 | 16a |

**Table 2.** Areas obtained in the IR spectra of the 1 nM pyridine solution on the twelve synthesized supports.

| Band (cm$^{-1}$) | Py | 5 mM | | | | 10 mM | | | | 20 mM | | | |
|---|---|---|---|---|---|---|---|---|---|---|---|---|---|
| | | 1 min | 2 min | 5 min | 10 min | 1 min | 2 min | 5 min | 10 min | 1 min | 2 min | 5 min | 10 min |
| 3301 | 40.61 ± 3.21 | 79.05± 12.62 | 97.63 ± 10.04 | 77.78 ± 5.26 | 75.71 ± 5.26 | 65.76 ± 2.96 | 74.56 ± 14.33 | 105.73 ± 0.74 | 111.23 ± 29.25 | 62.23 ± 13.57 | 83.67 ± 2.46 | 85.65 ± 9.82 | 91.95 ± 10.88 |
| 1636 | 14.68 ± 1.53 | 28.93 ± 6.71 | 32.35 ± 3.36 | 34.1 ± 12.92 | 24.81 ± 1.68 | 20.68 ± 0.62 | 28.38 ± 0.29 | 32.55 ± 4.99 | 36.28 ± 9.35 | 20.90 ± 4.39 | 30.56 ± 3.16 | 31.82 ± 3.72 | 32.61 ± 3.12 |

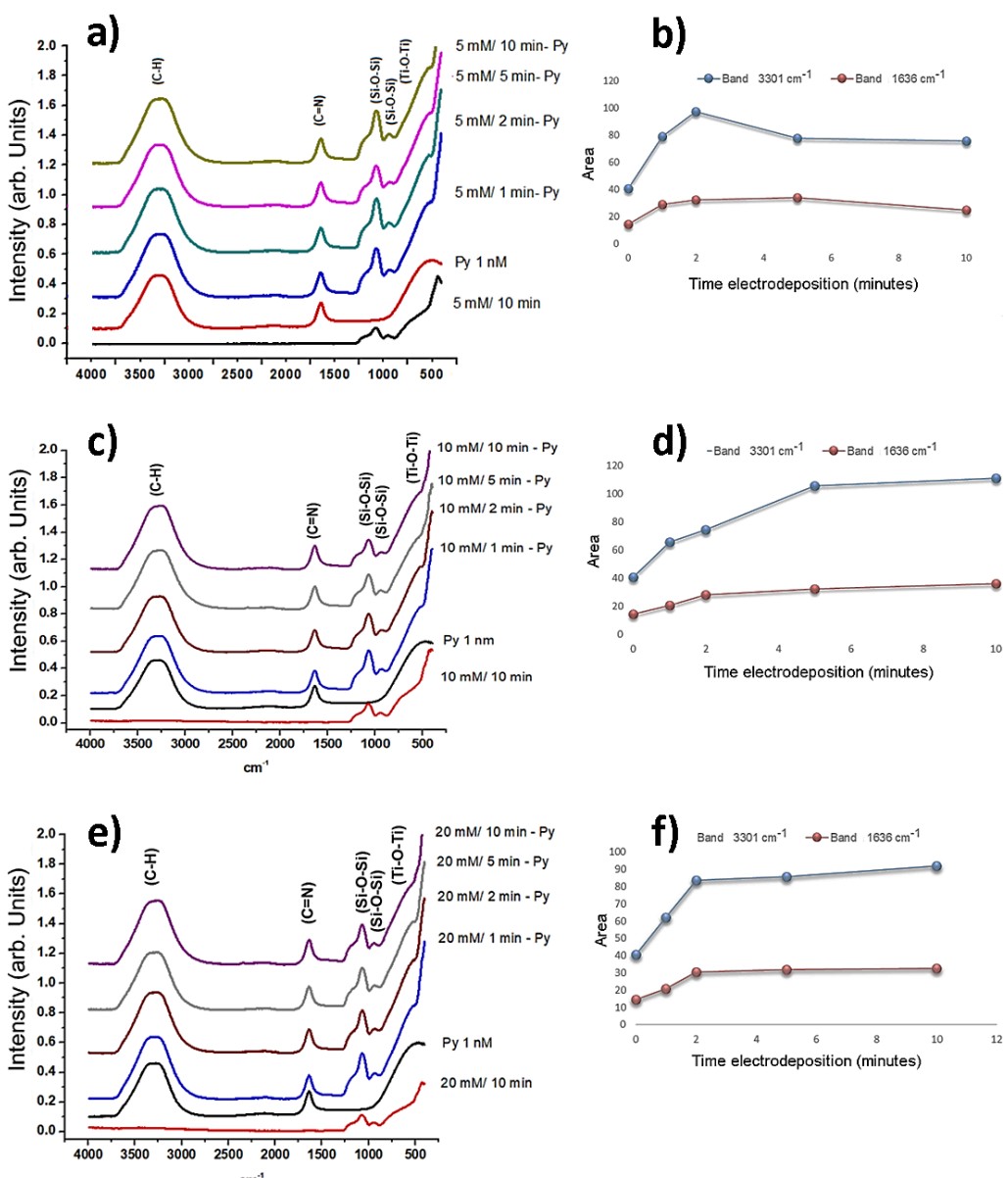

**Figure 7.** Infrared spectra of 1 nM pyridine on the supports doped at different times and concentration: (**a**,**b**) AgNO₃ at concentration of 5 mM (at 1, 2, 5 and 10 min); (**c**,**d**) AgNO₃ at concentration of 10 mM (at 1, 2, 5 and 10 min); (**e**,**f**) AgNO₃ at concentration of 20 mM (at 1, 2, 5 and 10 min). Note: pyridine solution and the supports alone are annexed.

In order to support the SEIRAS effect of the synthesized support, a crystal violet (CV) solution at a concentration of 1 mM was used as an additional analyte. It was found that there was also amplification, as shown in Figure 8. In addition, the same bands as in the pyridine solution, located at 3301 and 1636 cm$^{-1}$ for water, were observed. The bands belonging to the silica-titania support located at 1080, 952, 742, and 430 cm$^{-1}$ were also present. Due to the low concentration of both solutions, the observed amplified bands were found to be due to interaction with water.

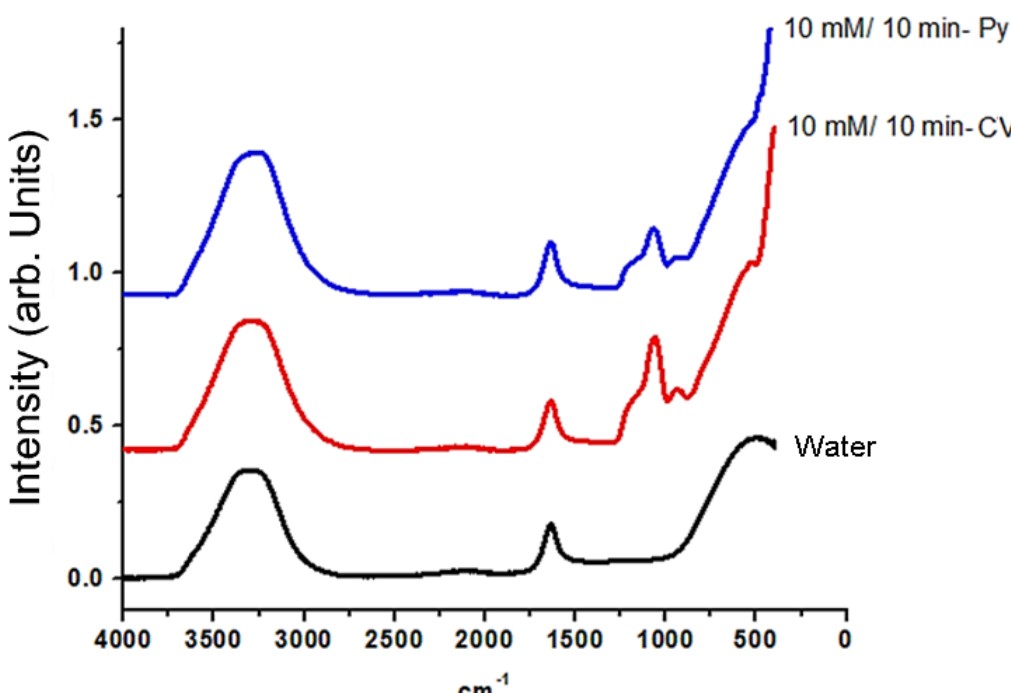

**Figure 8.** Infrared spectra of pyridine (1 nm) and crystal violet (CV) on the support doped for 10 min in a concentration of 10 mM AgNO$_3$, and water.

The amplification factor (*AF*) of the most intense bands in the *IR* spectra was calculated according to Equation (1) [27]:

$$AF = \frac{I_{SEIRAS}/I_{IR-R}}{C_{IR-R}/C_{SEIRAS}} \tag{1}$$

where $I_{SEIRAS}$ is the intensity of the band obtained from the pyridine on the support, $I_{IR-R}$ is the intensity of the band from the pyridine, $C_{IR-R}$ and $C_{SEIRAS}$ is the concentration of the reference and amplified solutions. Higher amplification was observed in the bands located at 3301 and 1636 cm$^{-1}$, in the twelve supports according to results in Table 3. The greatest effect occurred in the support doped for 10 min in a concentration of 10 mM AgNO$_3$ with an amplification factor equal to 2.74 and 2.47, followed by the support doped for 5 min in the same concentration. In these supports, large silver crystals predominate over small particles and dendrites, indicating that larger silver particles are required under the wavelengths of the infrared radiation beam for the formation of the Resonance Plasmon. The amplification was repeated in the concentrated pyridine and the crystal violet with AF of 2.01 and 1.7 for 3301 and 1636 cm$^{-1}$ bands. When the study was performed using concentrated pyridine the amplification factor differs approximately by a factor of about 10. The ATR-FTIR spectroscopy involves passing a beam of infrared radiation through a transparent glass to the IR and high refractive index, where the sample is placed. The incident IR beam is reflected several times and the surface of the sample absorbs part of the radiation at characteristic frequencies, with surface resonance plasmon occurring when a polarized light is directed from a prism with a high refractive index to a metal layer with a lower refractive index giving the amplification in the intensity of the pyridine bands.

**Table 3.** Amplification factor obtained from the two bands observed in the pyridine spectrum by FTIR.

| Band (cm$^{-1}$) | 5 mM | | | | 10 mM | | | | 20 mM | | | |
|---|---|---|---|---|---|---|---|---|---|---|---|---|
| | 1 min | 2 min | 5 min | 10 min | 1 min | 2 min | 5 min | 10 min | 1 min | 2 min | 5 min | 10 min |
| 3301 | 1.95 | 2.40 | 1.91 | 1.86 | 1.62 | 1.84 | 2.60 | 2.74 | 1.53 | 2.06 | 2.11 | 2.26 |
| 1636 | 1.97 | 2.20 | 2.32 | 1.69 | 1.41 | 1.93 | 2.22 | 2.47 | 1.42 | 2.08 | 2.17 | 2.22 |

### 3.2. Evaluation of Signal Amplification in Raman Spectroscopy

Figure 9 shows the Raman Spectrum obtained from the silica-titania-silver support, where four prominent bands are observed, at 148, 397, 517, and 639 cm$^{-1}$, which belong to the vibration modes Eg, B1g, A1g and B1g, and Eg of Anatase, which are active to Raman. The spectrum of the solution Pyridine 1 nM shows a small band at 1644 cm$^{-1}$ which was assigned to the stretching vibrations in the plane of C=C and C=N. The following four spectra belong to the pyridine solution, on each of the supports with silver at 1, 2, 5, and 10 min. In the support, the bands observed only correspond to the anatase. In the following supports with silver electrodeposition, it was possible to identify how the signal produced by pyridine was greater than the silica-titania fibers according with Figure 9a. On the electrodeposited support for two minutes, bands are shown at 148, 400, 529, and 645 cm$^{-1}$, belonging to the titania support; and bands at 703, 755, 826, 923, 981, 1240, 1292, 1363, 1408, 1466, 1531, 1589, and 1634 cm$^{-1}$, corresponding to the normal modes of vibration of pyridine: 16a, 10b, 10a, 16b, 5, 1, 3, 6a + 10a, 10a, 19b, 19a, 6a, 8a + 8b, 18b, respectively (see Figure 9a). The support doped for five minutes, on the other hand, showed a lower number of bands, located at 148, 406, 522, and 645 cm$^{-1}$, assigned to the normal vibrations of the anatase, at 1369, 1408, 1454, 1525, and 1589 cm$^{-1}$, corresponding to the normal vibration modes 10a, 19b, 19a, 6a and 8a + 8b of pyridine. The support doped for 10 min presented a lower number of bands corresponding to pyridine, located at 1232, 1283, 1363, 1411, 1459, 1590, and 1633 cm$^{-1}$ (Figure 9b), which are assigned to the vibrational modes 3, 6a + 10a, 10a, 19b, 19a, 8a + 8b, and 18b, respectively. Figure 9c,e shows similarly in the bands for the spectra obtained from the pyridine solution on each of the supports doped at different times in a 10 and 20 mM AgNO$_3$ solution, respectively. The bands observed in the spectra are summarized in Marked differences in the spectra are further definition of the bands on the doped supports for two minutes at a concentration of 5 mM AgNO$_3$, and at one minute at 10 mM of AgNO$_3$. The highest intensity of the bands was observed in the doped supports for two and five minutes in the concentration of 5 mM AgNO$_3$, in the support treated for 10 min in a concentration of 10 mM AgNO$_3$, and for 2 min in the concentration of 20 mM AgNO$_3$. The amplification of the bands was determined by calculating the area of the bands, and the results are shown in Figure 9b,d,f. In the case of the supports doped in a concentration of 5 mM AgNO$_3$, the highest signal amplification occurred in the support treated for five minutes, followed by the support doped for two minutes. The support with the highest SERS effect was obtained at 10 mM of AgNO$_3$ electroplated for ten minutes. Finally, for the supports developed at 20 mM, the greatest SERS effect was two minutes. The bands observed in the spectra are summarized in Table 4, with the correspondence of each band to the vibrational mode of titania and pyridine.

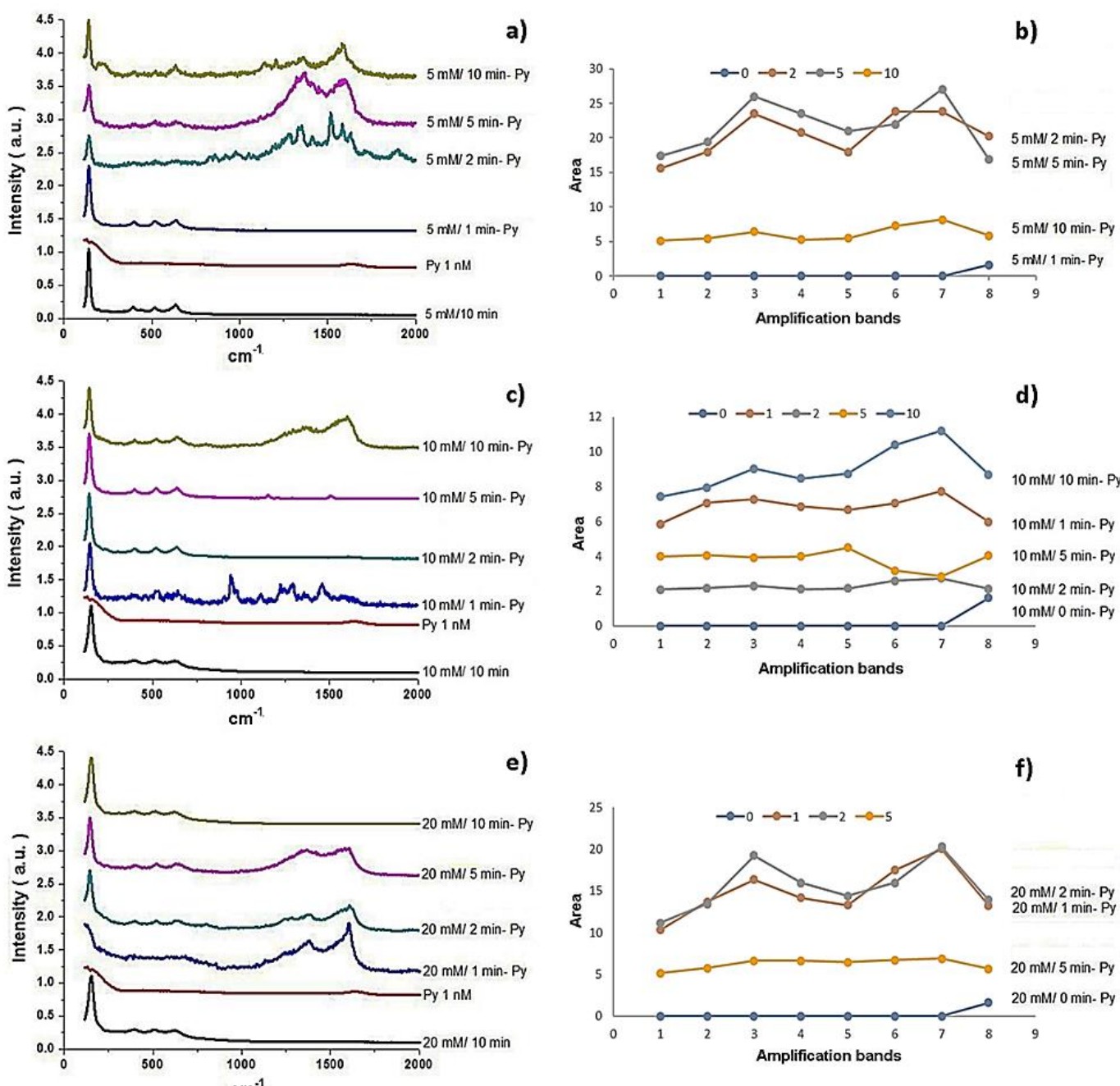

**Figure 9.** Raman spectra of 1 nM pyridine on the supports doped at different times and concentration: (**a,b**) AgNO₃ at concentration of 5 mM (at 1, 2, 5 and 10 min); (**c,d**) AgNO₃ at concentration of 10 mM (at 1, 2, 5 and 10 min); (**e,f**) AgNO₃ at concentration of 20 mM (at 1, 2, 5 and 10 min). Note: pyridine solution and the supports alone are annexed.

**Table 4.** Bands of Raman spectrum of the pyridine solution on the different $SiO_2$–$TiO_2$–Ag fiber supports, the bands observed in a wave number less than 645 $cm^{-1}$ belong to the anatase crystal structure of titania, the rest belong to pyridine. Cells marked with "X" indicate the presence of the band [3,30,31].

| Band ($cm^{-1}$) | AgNO$_3$ 5 mM Time (min) | | | | AgNO$_3$ 5 mM Time (min) | | | | AgNO$_3$ 5 mM Time (min) | | | | Vibrational Modes |
|---|---|---|---|---|---|---|---|---|---|---|---|---|---|
| | 1 | 2 | 5 | 10 | 1 | 2 | 5 | 10 | 1 | 2 | 5 | 10 | |
| 1634 | - | X | X | X | X | X | X | X | X | X | X | - | $18_b$ |
| 1589 | - | X | X | X | X | X | X | X | X | X | X | - | $8_a + 8_b$ |
| 1531 | - | X | X | X | X | X | X | X | X | X | X | - | $6_a$ |
| 1466 | - | X | X | X | X | X | X | X | X | X | X | - | $19_a$ |
| 1408 | - | X | X | X | X | X | X | X | X | X | X | - | $19_b$ |
| 1363 | - | X | X | X | X | X | X | X | X | X | X | - | $10_a$ |
| 1292 | - | X | X | X | X | X | X | X | X | X | X | - | $6_a + 10_a$ |
| 1240 | - | X | X | X | X | X | X | X | X | X | X | - | 3 |
| 981 | - | X | X | X | X | X | X | X | X | X | X | - | 1 |
| 923 | - | X | X | X | X | X | X | X | X | X | X | - | 5 |
| 826 | - | X | X | X | X | X | X | X | X | X | X | - | $16_b$ |
| 755 | - | X | X | X | X | X | X | X | X | X | X | - | $10_a$ |
| 703 | - | X | X | X | X | X | X | X | X | X | X | - | $16_a$ |
| 645 | X | X | X | X | X | X | X | X | X | X | X | X | $E_g$ |
| 522 | X | X | X | X | X | X | X | X | X | X | X | X | $A_{1g}$ y $B_{1g}$ |
| 406 | X | X | X | X | X | X | X | X | X | X | X | X | $B_{1g}$ |
| 148 | X | X | X | X | X | X | X | X | X | X | X | X | $E_g$ |

The amplification factor (*AF*) of the most intense Bands in the Raman spectra was calculated according to Equation (2):

$$AF = \frac{I_{SERS}/I_{Raman-R}}{C_{Raman-R}/C_{SERS}} \tag{2}$$

where $I_{SERS}$ is the intensity of the band obtained from the pyridine on the support, $I_{Raman-R}$ is the intensity of the band from the pyridine, $C_{Raman-R}$ and $C_{SERS}$ is the concentration of the reference and amplified solutions. The results are shown in Tables 5 and 6, data reflected higher amplification for the support at 5 mM AgNO$_3$ and 5 min.

**Table 5.** Amplification factor obtained from the intense bands observed in the pyridine Raman spectrum.

| Band ($cm^{-1}$) | AgNO$_3$ 5 mM Time (min) | | | | AgNO$_3$ 10 mM Time (min) | | | | AgNO$_3$ 20 mM Time (min) | | | |
|---|---|---|---|---|---|---|---|---|---|---|---|---|
| | 1 | 2 | 5 | 10 | 1 | 2 | 5 | 10 | 1 | 2 | 5 | 10 |
| 1634 | - | 12.54 | 10.43 | 3.62 | 3.71 | 1.32 | 2.50 | 5.39 | 8.22 | 8.62 | 3.52 | - |
| 1589 | - | 23.80 | 27.03 | 8.15 | 7.73 | 2.72 | 2.84 | 11.20 | 20.02 | 20.28 | 6.89 | - |
| 1531 | - | 24.76 | 21.97 | 7.27 | 7.05 | 2.58 | 3.18 | 10.39 | 17.49 | 15.94 | 6.71 | - |
| 1466 | - | 17.95 | 20.97 | 5.46 | 6.66 | 2.14 | 4.50 | 8.74 | 13.29 | 14.38 | 6.46 | - |
| 1408 | - | 20.75 | 23.48 | 5.26 | 6.85 | 2.10 | 4.00 | 8.46 | 14.19 | 15.97 | 6.64 | - |
| 1363 | - | 23.49 | 25.97 | 6.41 | 7.27 | 2.30 | 3.92 | 9.02 | 16.34 | 19.22 | 6.62 | - |
| 1292 | - | 17.96 | 19.36 | 5.42 | 7.07 | 2.17 | 4.04 | 7.94 | 13.71 | 13.44 | 5.75 | - |
| 1249 | - | 15.59 | 17.39 | 5.09 | 5.85 | 2.07 | 3.99 | 7.43 | 10.36 | 11.15 | 5.15 | - |

**Table 6.** Amplification factor for the bands observed in the Raman spectrum of the 1 mM violet crystal on the doped support for 5 min in a concentration of 5 mM $AgNO_3$.

| Band (cm$^{-1}$) | Amplification Factor | Vibrational Modes |
|---|---|---|
| 1597 | 25.26 | (C–C) $\upsilon_{s\ (Ring)}$ |
| 1547 | 18.12 | ($C_{ring}$–N) $\upsilon_s$ |
| 1423 | 16.45 | ($C_{center}$–C) $\upsilon_s$ |
| 1359 | 17.13 | (C–H) $_{\delta\ (Ring)}$ |
| 1190 | 12.76 | (C–C) $\upsilon_{s\ (Ring)}$ |
| 929 | 10.81 | (C–H) $_{\gamma\ (Ring)}$ |
| 810 | 9.77 | (C–H) $_{\gamma\ (Ring)}$ |
| 778 | 9.23 | (C–N–C) $_\delta$ |
| 513 | 10.54 | (C–H) $_{\gamma\ (Ring)}$ |
| 456 | 7.73 | (C–$C_{Center}$–C) $_\delta$ |
| 326 | 7.80 | (C–C) $\upsilon_{s\ (Ring)}$ |

Subsequently, the SERS effect was analyzed again on the support that showed the highest amplification (5 min/5 mM $AgNO_3$) using a crystal violet solution with a concentration of 1 mM as the analyte, yielding the spectra observed in Figure 10. The amplification of the bands appeared between 500 and 1300 cm$^{-1}$. The bands located at 1597 and 929 cm$^{-1}$ indicated the vibration of the C–C bonds of the rings, while those at 1547 and 513 cm$^{-1}$ belong to the vibration of the C–N bonds; the vibrations of the bonds between the rings and the central carbon are manifested at 1359 and 326 cm$^{-1}$, and the bands located at 1190, 810, 778 and 445 cm$^{-1}$ are due for the vibrations of the C–H bonds of the ring; the spectra bands are presented in Table 6. When evaluating the SERS effect of the twelve supports using pyridine as an analyte, the amplification effect was detected. The pyridine bands had a higher amplification in the doped support for five minutes at a concentration of 5 Mm, where the silver formed defined dendrites with a distance between the branches of $100 \pm 9$ nm, that grew along the silica-titania fibers; it was these small separations that generated hot spots that led to the amplification of the signals. This effect was not observed after two minutes of treatment, since there were not enough dendrites in the fibers and their morphology was not fully defined at five minutes in the support. In the same way, the amplification phenomenon was diminished after ten minutes of treatment, because the dendrites fused to form larger particles and the hot spots were lost. The behavior was repeated with the doped supports at concentrations of 10 and 20 mM. The highest amplification occurred in the doped support for five minutes in a concentration of 5 mM $AgNO_3$, because it was the only one in which most of the dendrites grew along the fibers, with the appropriate structure, and not in small random agglomerates of dendrites as in the rest of the supports. The amplification factor reached a value of 27.03 in the most intense band at 1589 cm$^{-1}$, compared to previous studies, where the band with the highest intensity reached an amplification factor of 13.6 in the support doped for 10 min in a concentration of 10 mM $AgNO_3$.

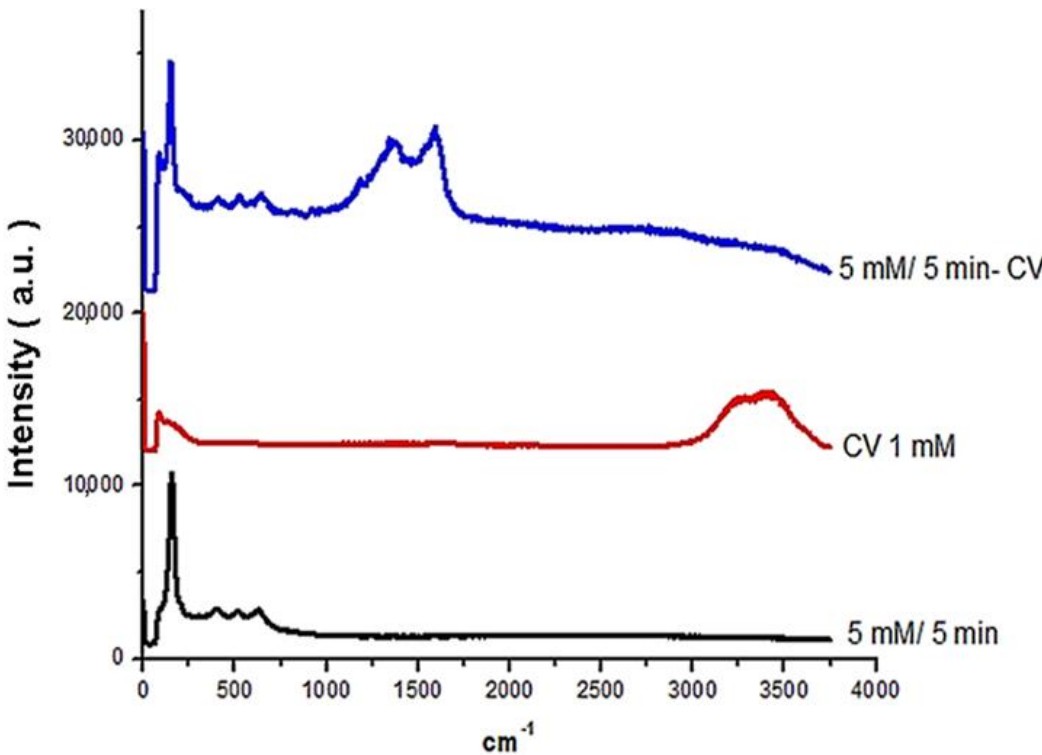

**Figure 10.** Raman spectra of crystal violet (CV) on the support doped for 5 min at 10 mM of AgNO$_3$.

### 4. Conclusions

Homogeneous, smooth, and defect-free silica-titania fibers with an average diameter of 150 ± 8 nm were developed using sol-gel and electrospinning techniques. Twelve different supports were developed by varying the concentration of the AgNO$_3$ solution and electrodeposition time. The silver deposited on the silica-titania fibers presented various structures, depending on the concentration and the treatment time, such as particles without defined faces, dendrites, crystals with hexagonal faces, and almost spherical crystals. Silica-titania-Ag fibers with the ability to amplify signals in ATR infrared spectroscopy were successfully developed. The support doped for 10 min in a concentration of 10 mM AgNO$_3$ presented the greatest amplification due to the large size of the crystals formed, which was optimal for the formation of the resonance plasmon under the incident wavelengths. The amplification effect in infrared spectroscopy for the supports was reproducible using concentrated pyridine as a sample. The metal surface in silica-titania fibers contributes considerably to the mechanism of amplification, and the effective range of signal amplification was found to be in the vicinity of concentrations of 1 nM.

The support doped for five minutes at a concentration of 5 mM AgNO$_3$ showed the greatest amplification in Raman spectroscopy, where silver formed defined dendrites with a distance between the branches of 100 ± 9 nm, which grew along the silica-titania fibers, forming hot spots that led to the amplification of the signals. The SERS effect was reproduced using a 1 mM crystal violet solution as the analyte. The highest amplification factor reached by the SERS technique was 27.03 in the band formed by the 8a + 8b vibrational mode of pyridine and 25.26 in the band located at 1597 cm$^{-1}$.

**Author Contributions:** Conceptualization, S.Y.R.-L.; methodology, S.Y.R.-L., R.F. and B.S.C.-R.; formal analysis, S.Y.R.-L. and B.S.C.-R.; investigation, S.Y.R.-L. and B.S.C.-R.; resources, S.Y.R.-L.; data curation, S.Y.R.-L. and B.S.C.-R.; writing—original draft preparation, S.Y.R.-L.; writing—review and editing, S.Y.R.-L.; visualization, S.Y.R.-L.; supervision, S.Y.R.-L. and R.F.; project administration, S.Y.R.-L.; funding acquisition, S.Y.R.-L. All authors have read and agreed to the published version of the manuscript.

**Funding:** This research received no external funding.

**Institutional Review Board Statement:** Not applicable.

**Informed Consent Statement:** Not applicable for studies not involving humans.

**Data Availability Statement:** The data used to support the findings of this study are available from the corresponding author upon request.

**Acknowledgments:** Thanks to PRODEP, Universidad Autónoma de Ciudad Juárez and CONACYT for supporting this investigation.

**Conflicts of Interest:** The authors declare that there are no conflict of interest regarding the publication of this paper.

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
