# Peer review of "Surface Enhanced Infrared Absorption Studies of SiO2–TiO2–Ag Nanofibers: Effect of Silver Electrodeposition Time on the Amplification of Signals"

_crystals, doi:10.3390/cryst11050563_

Round 1

Reviewer 1 Report

Manuscript Review

Manuscript ID: crystals-1175338

Title: Surface enhanced infrared absorption studies in SiO2-TiO2-Ag nanofibers: Effect of silver electrodeposition time in the amplification of signals.

The authors describe an interesting study of the effect of the used silica-titania-silver nanosubstrate for the SERS and SEIRAS signal enhancement. The authors designed 12 different active substrates using sol-gel and electrospinning techniques. The paper is well-presented, good length and the results supports the conclusions in a convincing way. The performed analysis was based on the latest literature reports. This is an interesting work that gives new results and deserves publication after consideration of several minor remarks:

  1. Abstract: Please specify and underline why this study is crucial and what it can contribute to the development of research in this field. Is it just about overcoming the high cost, low stability and low compatibility of the materials already available?
  2. Introduction part: Surface selection rules (invited by Moskovits, Creighton, Castro, Osawa and Lombardi), and the enhancement mechanisms (chemical and electromagnetic) in SERS and SEIRA techniques should be described. Therefore I suggest to expand a little bit the Introduction section to better introduce the Reader to the topic discussed in the publication.
  3. Experimental part: Could you provide more details about FTIR/SEIRAS and RS/SERS measurements conditions, spectra processing parameters?
  4. Results and discussion: The observed spectral differences between the classical FTIR and strengthened spectral features (Figure 6) are crucial to draw correct conclusions. Could authors present spectra more clearly? The appropriate normalization and substantially baseline correction should be applied. Perhaps the authors should consider split the spectrum into high and low frequency ranges to make the bands more visible.
  5. Results and discussion: Table 1: The authors introduce the Wilson numbering scheme to list the appropriate band assignments. Please introduce the appropriate reference into the reference list.
  6. Results and discussion: I suggest always presenting spectra in one convention - either from the highest frequency to the lowest or vice versa.

Author Response

  1. Abstract: Please specify and underline why this study is crucial and what it can contribute to the development of research in this field. Is it just about overcoming the high cost, low stability and low compatibility of the materials already available?

Response: New information is added.

  1. Introduction part: Surface selection rules (invited by Moskovits, Creighton, Castro, Osawa and Lombardi), and the enhancement mechanisms (chemical and electromagnetic) in SERS and SEIRA techniques should be described. Therefore I suggest to expand a little bit the Introduction section to better introduce the Reader to the topic discussed in the publication.

Response: New information is added.

  1. Experimental part: Could you provide more details about FTIR/SEIRAS and RS/SERS measurements conditions, spectra processing parameters?

Response: New information is added.

  1. Results and discussion: The observed spectral differences between the classical FTIR and strengthened spectral features (Figure 6) are crucial to draw correct conclusions. Could authors present spectra more clearly? The appropriate normalization and substantially baseline correction should be applied. Perhaps the authors should consider split the spectrum into high and low frequency ranges to make the bands more visible.

Response: New information is added. All spectra have a normalization and baseline correction. The calculus was made with software.

  1. Results and discussion: Table 1: The authors introduce the Wilson numbering scheme to list the appropriate band assignments. Please introduce the appropriate reference into the reference list.

Response: New information is added.

  1. Results and discussion: I suggest always presenting spectra in one convention - either from the highest frequency to the lowest or vice versa.

  1. Response: IR and Raman spectra are reported according standard convention.

Reviewer 2 Report

The work is interesting and huge amount of of results is summarized. There are some minor corrections and comments:

  • when first writing an abbreviation in the text (introduction) it should be explained
  • line 95 (will be) - so far past tense was used
  • line 123-124 - sentence is not clear to me
  • figure 1 needs larger labels
  • line 55 (vibration of the pyridine - weren't you just looking at the fibers without probe molecule?
  • figure 2 needs larger labels
  • better scale bars on the SEM images required
  • line 276 - 701.676 - use comma instead
  • Table 1 - is it possible to construct smaller table for such a few results
  • Table 2 - what do you mean by area?
  • Figure 8 - it is written VC instrad of CV
  • There are few sentences that are too long and hard to understand (for example - lines 114-117; lines 431-436; lines 469-471 (should be split in two sentences)).

Author Response

The work is interesting and huge amount of of results is summarized. There are some minor corrections and comments:

  • when first writing an abbreviation in the text (introduction) it should be explained
  • Response: Abbreviation was explained.
  •  
  • line 95 (will be) - so far past tense was used

  • Response: Text is corrected.

  • line 123-124 - sentence is not clear to me

  • Response: Text is corrected.
  •  
  • figure 1 needs larger labels

Response: Figure is corrected

  • line 55 (vibration of the pyridine - weren't you just looking at the fibers without probe molecule?
  • figure 2 needs larger labels
  • Response: Figure is corrected
  •  
  • better scale bars on the SEM images required
  • Response: Figure is corrected
  •  
  • line 276 - 701.676 - use comma instead
  • Response: It is correct
  • Table 1 - is it possible to construct smaller table for such a few results
  • Response: It is correct
  •  
  • Table 2 - what do you mean by area?
  • Figure 8 - it is written VC instrad of CV
  • Text is corrected.
  • There are few sentences that are too long and hard to understand (for example - lines 114-117; lines 431-436; lines 469-471 (should be split in two sentences)).
  • Text is corrected.